

# Examining the influence of picture format on children's naming responses

Naroa Martínez and Helena Matute

Departamento de Fundamentos y Métodos de la Psicología, Universidad de Deusto, Bilbao, Spain

## ABSTRACT

Digital photography has facilitated the use of more ecological stimuli than line drawings as experimental stimuli. However, there is lack of evidence regarding the effect of the picture format on children's naming agreement. The present work investigated whether the format of presentation of the pictures (line drawing or photograph) affects naming task performance in children. Two naming task experiments are reported using 106 concepts depicted both as a photograph and as a matched drawing delineated directly from the photograph. Thirty-eight and thirty-four Spanish-speaking children from 8 to 10 years old participated in Experiment 1 and Experiment 2, respectively. We examined name agreement measures ($H$ index, percentage of modal name, and alternative responses) and subjective scales (familiarity and visual complexity). The results revealed a significant main effect of format in all of the variables except for familiarity, indicating better name agreement indices and higher visual complexity values for the photograph format than for the line drawing format. Additionally, line drawings were more likely to produce alternative incorrect names. The implications of these findings for psychoeducational research and practice are discussed.

## INTRODUCTION

Pictures play an important role in psychoeducational assessment, intervention, and research. For instance, picture naming is a very frequently used task because it allows for exploring various cognitive processes such as perceptual processing, activation of semantic information, lexical selection, name retrieval, and motor planning (see *Bonin et al., 2015*; *Humphreys & Riddoch, 2006*; *Levelt, Roelofs & Meyer, 1999*; *Riddoch & Humphreys, 2001*; *Roelofs & Ferreira, in press*). In children, pictures—as opposed to written stimuli—could be the only option when conducting researching with pre-reader kindergarten pupils. Pictures might also be considered more age-appropriate for primary school children, since they exclude possible effects derived from reading skills (see *Perfetti, Finger & Hogaboam, 1978*).

One important line of research, which has provided well-controlled stimuli for picture naming tasks, consists of normative studies. Norms offer information on the variables of central relevance that influence naming performance. A pioneering normative study is the one reported by *Snodgrass & Vanderwart (1980)*. This consisted of 260 black-and-white line drawings with norms for name agreement, image agreement (or the degree of

Corresponding author
Naroa Martínez,
naroa.martinez@deusto.es

agreement between the mental image and the picture), familiarity, and visual complexity of English-speaking adults. The results showed that pictures were named more accurately when a particular word represented by a picture was frequent and familiar, when the picture had low subjective visual complexity, and when the image agreement was high. Since then, vast literature has emerged in which these line drawing sets have been extended and adapted to numerous languages for use with adults (e.g., *Sanfeliu & Fernandez, 1996*; in Castilian Spanish; *Weekes et al., 2007*; in Chinese; *Alario & Ferrand, 1999*; in French; *Nisi, Longoni & Snodgrass, 2000*; in Italian; *Pind et al., 2000*; in Icelandic; *Nishimoto et al., 2005*; in Japanese; *Van Schagen et al., 1983*; in Dutch), and in children (e.g., *Wang, Chen & Zhu, 2014*; in Chinese; *Piñeiro, Manzano & Reigosa, 1999*; in Cuban Spanish; *Berman et al., 1989*; *Cycowicz et al., 1997*; in English; *Cannard et al., 2006*; in French; *D'Amico, Devescovi & Bates, 2001*; in Italian; *Pompéia, Miranda & Bueno, 2001*; in Portuguese). Some other works have been dedicated to collect norms on the colored and textured version of the Snodgrass and Vanderwart set (*Rossion & Pourtois, 2004*; in English; *Tsaparina, Bonin & Méot, 2011*; in Russian; *Raman, Raman & Mertan, 2014*; in Turkish; *Dimitropoulou et al., 2009*; in Greek; *Bakhtiar, Nilipour & Weekes, 2013*; in Persian). It is worth mentioning that the majority of the normative studies of picture naming have been based on the same set of stimuli, the *Snodgrass & Vanderwart (1980)* set, or its corresponding colorized version (*Rossion & Pourtois, 2004*). Very few studies have used a different set of line drawings. These are, for example, the set offered by *Bonin et al. (2003)* in French; the Protocole Européen de Dénomination Orale d'Images (PEDOI, *Kremin et al., 2003*) in Dutch, English, German, French, Italian, Russian, Swedish, and Spanish; or the Multilingual Picture databank (MultiPic, *Duñabeitia et al., 2018*) in Spanish, British English, German, Italian, French and Dutch.

In addition to drawing format datasets, technological development has facilitated the use of digital photographs as experimental stimuli. Moreover, the standardization of photographic sets for adults has progressively increased in recent years (see, for example, *Adlington, Laws & Gale, 2009*; *Russo et al., 2018*; in English; *Brodeur et al., 2010*; *Brodeur, Guérard & Bouras, 2014*; in English and French; *Saryazdi et al., 2018*; in Turkish; *Shao & Stiegert, 2016*; in Dutch; *Moreno-Martínez & Montoro, 2012*; in Spanish; *Navarrete et al., 2019*; in Italian). Thus, the need for more ecological stimuli than those provided by line drawings has begun to be highlighted. However, to the best of our knowledge, no standardized set of photographs with norms for children has yet been published. For this reason, we recently created a new bank with both photographs and matched line drawings with norms for children (N Martínez, H Matute & E Goikoetxea, 2019, unpublished data).

In the present research, we aim to test whether significant differences exist as a function of picture format (i.e., photographs vs. drawings) on picture naming tasks. Both line drawings and photographs have different characteristics that should affect object recognition and naming responses. Line drawings are schematic, simple, and prototypical representations of concepts whilst photographs offer a realistic representation including color and surface details such as texture, along with information about volume, brightness, and shade. Below we discuss studies with adults that have revealed a picture format effect in picture naming. However, similar studies are scarce for children, partly because that normative

data of photographic sets have not yet been collected. We will also present evidence of the photograph facilitation effect when compared with the use of line drawings in children's object recognition, reported with the use of tasks other than picture naming.

Some studies with adults have examined the effect of picture format in a picture naming task by comparing drawings and photographs of different sets of stimuli (*O'Sullivan et al., 2012*; *Shao & Stiegert, 2016*) or using the same set of stimuli (*Salmon, Matheson & McMullen, 2014*). In addition, a few studies matched also the shape, scale, and orientation features between drawings and photographs (*Price & Humphreys, 1989*; *Brodie, Wallace & Sharrat, 1991*; *Saryazdi et al., 2018*). Of those, the image types and the naming variables examined differed between the different studies. For example, *Price & Humphreys (1989)* examined the effect of picture features on naming accuracy (the percentage of error) and reaction time in three experiments by comparing five different picture formats (correct color photograph, black-and-white photographs, correct color line-drawings, black-and-white line drawings, and incongruent color line drawings). Among the main findings, they found that both the correct color and the photographic details improved naming accuracy and reduced naming reaction time. *Brodie, Wallace & Sharrat (1991)* found a progressive decrease in naming latencies from line drawings to grayscale photographs to color photographs. They stated that surface details presented in photographs (e.g., texture and three-dimensional cues) could facilitate recognition and this information is typically missing in line drawings. Recently, *Saryazdi et al. (2018)* explored differences between cliparts and colored photographs of 225 objects in several measures (modal name and verb agreement measures, picture–name agreement, familiarity, visual complexity, and image agreement). Cliparts were sophisticated colored drawings created by editing photographs. They observed analogous results across cliparts and photographs. However, even with these two similar types of stimuli, there were significantly higher ratings of verb agreement and picture-name agreement for photographs than for cliparts, although these differences were small in magnitude. The authors discussed the subtle differences observed in terms of a visual iconicity effect, that is, the perceptual similarity between the picture and its referent. In this regard, photographs make the referential relationship more transparent, which could help to transfer information between the picture and the real object. Taking into account the results of the above studies, it seems that the greater the difference between the line drawing and the photograph, the greater the effect of image format on object recognition. That is, if the line drawings are not created directly from the photographs and do not incorporate color and surface details, it is possible to observe higher naming accuracy scores and faster reaction times in photographs than in line drawings.

In fact, it has been extensively studied how color and surface details affect object recognition in drawings and in photographs, separately. For example, it has been shown that color improves name agreement (*Rossion & Pourtois, 2004*), memorization (*Vernon & Lloyd-Jones, 2003*), and naming speed of objects (e.g., *Bonin et al., 2019*; *Rossion & Pourtois, 2004*) by comparing naming performance when using black-and-white, grayscale, and colored drawings; and identification and memorization of objects by comparing black-and-white photographs and colored photographs (e.g., *Lloyd-Jones & Nakabayashi, 2009*; *Uttl, Graf & Santacruz, 2006*). In a meta-analysis, *Bramão et al. (2011)* examined

the effect of color on object recognition, mainly using naming tasks, in 35 experiments involving 1,535 adult participants. They found a moderate effect of color on the recognition of the object in line drawings, photographs, and photographs without superficial details. A recent study conducted by *Bonin et al. (2019)* examined the effect of color and the role of the surface details in naming performance. This study compared written naming latencies of the same objects in black-and-white (*Snodgrass & Vanderwart, 1980*), grayscale and colored drawings (*Rossion & Pourtois, 2004*). They found that colored drawings yield shorter written naming latencies than grayscale drawings, and these, in turn, shorter latencies than black-and-white drawings. The inclusion of grayscale texture and shading without color did not reliably improve naming performance as indexed by name agreement scores, a result that agrees with *Rossion & Pourtois (2004)* findings.

The superiority of photographs over drawings has been shown in the engagement of manual exploration in 9-month-old infants (*Pierroutsakos & DeLoache, 2003*), imitative performance for a novel action on the basis of a picture-book interaction in 18-month-old infants (*Simcock & DeLoache, 2006*) and in success in matching real objects with pictures in 3-year-old children (*Callaghan, 2000*). With respect to visual iconicity, studies carried out with children from 1 to 3-years of age showed that they were better able to relate the image to the object when images were more iconic in comparison with images that were less realistic (*Ganea, Pickard & DeLoache, 2008*).

To summarize, line drawings and photographs both have different characteristics that affect naming performance in adults and imitative and matching performance in children. However, to the best of our knowledge, there is a lack of studies carried out with children to examine the effect of the picture format on picture naming tasks. The aim of the present study was to compare several variables (the name agreement scored according to the *H* index and the percentage of the modal name, the alternative names and unknown responses classified into different categories, familiarity, and visual complexity) in picture naming tasks with children using both line drawing and photographic stimuli. Thus, we explored the possible differences between the simplest and most schematic form of a set of pictures (line drawings) with a more ecological, visually iconic, and complete two-dimensional format of such pictures (photographs).

## EXPERIMENT 1
### Materials & methods
#### Participants
A total of 38 native Spanish-speaking children participated in the study: 17 were from the 3rd grade (53% girls, *M* age = 8 years 3 months, *SD* = 4.69 months) and 21 were from the 4th grade (43% girls, *M* age = 9 years 4 months, *SD* = 5.10 months). None of the participants had received a diagnosis of neurological damage or problems with speaking or language. Two additional children were excluded from the sample for not completing the task. All of the children attended a public school in Madrid that serves families with a middle-low socio-economic level. The written informed consents of the adults responsible for the children who participated in the study were collected, and all children agreed verbally to take part in the study.

The sample was selected according to the following considerations. Primary school children are a sector of the population that requires well-controlled visual stimuli and one with which pictures are widely used in educational materials and assessment instruments (e.g., test WISC-IV; *Wechsler, 2003*). In addition, data collection through a written naming task facilitates conducting research with large groups of participants efficiently (e.g., *Berman et al., 1989*; *Bonin et al., 2019*). It also reduces significantly the time and cost of the experiments with children, as long as children have the basic spelling skills that allow them to write words fluently. In this respect, 3er and 4th graders constitute a good sample because by the end of the second grade Spanish-speaking children achieve a basic level of spelling proficiency (*Defior, Jiménez-Fernández & Serrano, 2009*). In fact, a pilot study of this experiment was carried out with children 5–7 years-old. We decided to conduct the pilot study individually due to the considerable differences in writing speed among the participants. In addition, we observed that fatigue might be a problem, so we decided to split the tasks in three phases, which further lengthened the time of data collection. Moreover, a recent study by *Schmetz et al. (2018)* assessed how basic visual processes progress in 215 children from 4 to 14 years old and in 20 adults. The results showed that processing of surfaces reached maturity by the age of 9–10 years, processing of length and position by the age of 13–14 years, and orientation processing continues to improve beyond the age of 14 years.

### Ethics statement

The ethics committee of the University of Deusto approved the procedure of the present study (Ref: ETK-14/17-18).

### Materials

We used the PicPsy bank (N Martínez, H Matute & E Goikoetxea, 2019, unpublished data). This bank consists of 106 concepts. Each of them is depicted both as a photograph and as a matched drawing delineated directly from the photograph (see Fig. 1 for a sample of the stimuli). We selected half of the stimuli from the bank for Experiment 1 and the other half for Experiment 2. We decided to divide the bank in two different sets in order to prevent fatigue and loss of attention in children (*D'Amico, Devescovi & Bates, 2001*) because our presenting each concept in two different picture formats doubled the number of trials. Thus, 53 concepts, each depicted as both a photograph and a drawing (106 pictures in total) were used in this experiment.

In PicPsy (N Martínez, H Matute & E Goikoetxea, 2019, unpublished data), the concepts represented by the pictures were selected taking into account different psycholinguistic variables and subjective ratings (see Table 1). The psycholinguistic variables were: lexical frequency diversity according to Spanish dictionary of word frequency in children's writing (*Martínez & García, 2004*), and different length. The subjective ratings were: high familiarity indexes, high imagination indexes, high concreteness indexes according to a scale of 1–7 of ES-PAL, and the subjective age of acquisition under 8 years according to *Alonso, Fernandez & Díez (2015)*. We found no significant differences between the list of concepts used in Experiment 1 and those used in Experiment 2 in terms of the mentioned variables. Most of the concepts used in this experiment (70% approximately) overlapped

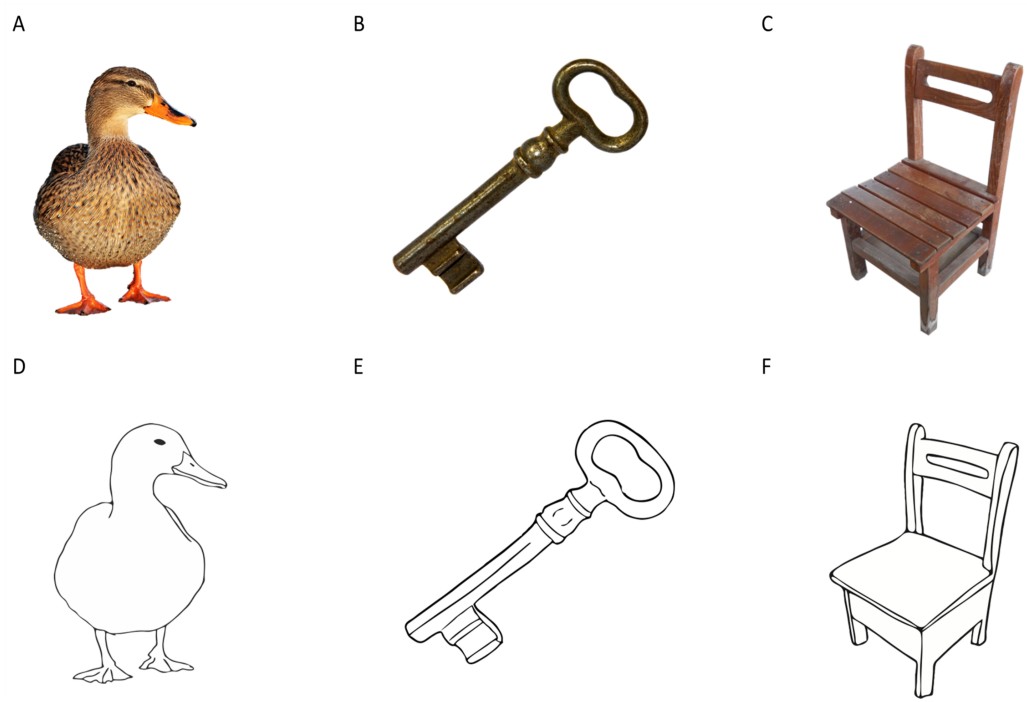

**Figure 1** **Examples of picture pairs in line drawing and photograph format.** The photographs were retrieved from https://pixabay.com under a CC0 license. Image credit: Naroa Martínez and Helena Matute.

**Table 1** **Descriptive statistics of the psycholinguistic variables of the stimuli in Experiment 1.**

| Variable | Database | *M* (*SD*) | Range |
|---|---|---|---|
| Linguistic variables | | | |
|     Lexical frequency | *Martínez & García (2004)* | 127.82 (237.68) | 2.51–1,580.43 |
|     Length | | 4.81 (1.66) | 3–11 |
| Subjective ratings | | | |
|     Familiarity | *Duchon et al. (2013)* | 6.12 (0.64) | 3.56–7 |
|     Imagination | *Duchon et al. (2013)* | 6.13 (0.51) | 4.57–6.85 |
|     Concreteness | *Duchon et al. (2013)* | 5.83 (0.64) | 3.74–6.77 |
|     Age of acquisition | *Alonso, Fernandez & Díez (2015)* | 4.23 (1.29) | 2.32–7.36 |

with those of *Goikoetxea (2000)* for 3rd and 4th graders, and corresponded to 25 different semantic categories, namely: animals, atmospheric phenomena, birds, buildings, clothing, feelings, flowers, fruits, furniture, geographical accidents and natural land formations, insects, kitchen utensils, light sources, mammals, parts of a house, parts of the human body, reading material, tools, trees, types of boats, types of fabrics, types of food, types of professions, types of relatives, units of weight.

All photographs in PicPsy (N Martínez, H Matute & E Goikoetxea, 2019, unpublished data) were downloaded from free databases (mostly from https://pixabay.com) under a CC0 public domain license. For editing the photographs, the procedure followed was

similar to that employed by *Brodeur et al. (2010)* and *Brodeur, Guérard & Bouras (2014)* using PowerPoint and CorelDraw (Corel Corp., Ottawa, Canada). The four editing steps were: (1) cut the object from the scene, (2) blur the words, (3) resize the image to fit within a frame of $500 \times 500$ pixels, and (4) in some images, arrows or other images were added to improve the representation of the concept. For the arrangement of the photographs, we adopted the following criteria based on those described previously in the work of *Snodgrass & Vanderwart (1980)*: (a) in the case of animals or parts of the body, approximately the same number of images were shown oriented to the right and to the left; (b) in the case of objects whose orientation upwards and downwards may vary (e.g., fork), the functional part was placed downwards with approximately the same number of photographs oriented to the right and to the left; and (c) fine and elongated objects were oriented with a 45° inclination.

The drawings in PicPsy (N Martínez, H Matute & E Goikoetxea, 2019, unpublished data) were prepared by one of the authors who delineated photographs by hand. After that, they were digitized to follow the same editing procedure as the photographs. Once edited, each line drawing was vectorized to achieve adequate resolution. Vectorization transforms a picture to vectors instead of pixels, which allows for the enlargement or reduction of the image to any size without modifying its high quality, due to its defined contours. The line drawings preserved the same scale, shape, and orientation as the corresponding photographs.

### Procedure

Each participant completed the picture-naming task in both formats (line drawing and photograph). The order of format presentation was counterbalanced. In order to control the potential effects of order and sequence in the repeated-measures design, children were randomly assigned to one of two different groups. One group was presented the set in line drawings first, and the other group was presented with the photographs first. No significant differences were found in the age of the counterbalancing groups, $t(36) = -0.64$, $p = .523$, $d = -0.21$.

All participants were evaluated in groups by one examiner trained in the administration of the task in a quiet room of the school at the beginning of the 2017–2018 year. Stimuli were presented, one by one, at the center of a computer screen. Each picture was preceded by a fixation point (+) for 500 ms, and remained on the screen for around 5,000 ms or until the participants responded. The sequence of presentation of the stimuli was assigned randomly but the sequence remained the same for each format order. The variables for analysis as well as their corresponding instructions, which were presented written and orally to the participants, were as follows:

*Picture naming.* Participants were asked to give a single name for each item by writing the first word that came to mind in order to name each picture. In cases where the name was unknown to them, they were instructed to write the initials DKN for "Don't Know the Name" (in Spanish NSP for No Sé la Palabra), DKO was used for "Don't Know the Object"

(in Spanish NSO for No Sé el Objeto), and TOT was used for "Tip-Of-the-Tongue" (in Spanish PDL for Punta De la Lengua).

*Familiarity.* Participants were asked to rate the familiarity of the word represented by the picture according to how often they daily interact with, hear, or think about the word on a scale from 1(a few times) to 5 (many times). Participants were asked to rate the word itself rather than the picture.

*Visual complexity.* Participants were asked to rate the visual complexity of the picture according to the number of details they thought the picture had on a scale from 1(few details) to 5 (many details).

Each participant completed three tasks (picture naming, familiarity rating, and visual complexity rating) in both formats (line drawing and photograph). Each child, therefore, responded to a total of 106 stimuli: 53 in line drawing and 53 in photographic format. These three tasks are those that are usually included in picture naming studies with children (e.g., *Cannard et al., 2006*; *Berman et al., 1989*; *Cycowicz et al., 1997*; *D'Amico, Devescovi & Bates, 2001*; *Pompéia, Miranda & Bueno, 2001*). However, variables such as image agreement and image variability are not usually included due to children's difficulty to understand these tasks, because they require children to form and judge mental images (see *Wang, Chen & Zhu, 2014*), in addition to handling many scales simultaneously.

All children responded on a sheet of paper created for this purpose. The sheets included numbered lines, one for each picture, where participants had to write down their answers for the naming task. Following each line, two scales were presented where they were asked to indicate with an *X* the corresponding value for the rating of familiarity and visual complexity. The scales of familiarity and visual complexity were adapted from those proposed by *Piñeiro, Manzano & Reigosa (1999)*. The values of the scale were complemented with a visual scale composed of squares of different sizes in order to facilitate the handling of the scales by the children. Thus, to represent the values 1, 3 and 5 of the scale three rectangles of sizes 1 × 1 cm, 1 × 3 cm and 1 × 5 cm were presented, all of which were gray added with a level of transparency of 50, 30, and 10%, respectively. During the session, the researchers gave prior instructions and two examples (not included in the set) were completed in order to allow each participant to become familiar with the task and the scales.

### Scoring

The responses of the participants were transcribed, and were corrected for spelling errors. Basic variants of the same name such as singular and plural forms (e.g., windows and window) were collapsed. For responses that included one or a second noun (e.g., frog or toad), the noun recovered second was excluded.

Different measures of name agreement were analyzed: the *H* index, the percentage of modal name responses, the percentage of alternative names in each category, and the percentage of unknown responses in each category.

The *H* index for each of the 53 drawings and each of the 53 photographs was analyzed. The *H* index is a statistic (*Shannon & Weaver, 1949*), which was introduced by *Snodgrass*

*& Vanderwart (1980)* as a name agreement index to reflect the dispersion of the name responses for each stimulus (drawing or photograph) and is calculated as follows:

$$H = \sum_{i=1}^{k} P_i \log 2(1/P_i),$$

where $k$ represents the number of different names given for each picture, and $P_i$ represents the proportion of participants who mentioned each valid name. Thus, an $H$ value of 0 indicates that only one name has been given for an image. The increase in the $H$ values indicates an increase in the dispersion of the responses.

In addition, the percentage of participants who responded with the modal name was calculated (name given by most subjects in the sample). Although both the $H$ index and the percentage of modal name responses are measures of the naming agreement, the percentage only indicates how dominant the modal name is in the sample while the $H$ index is sensitive to how widely the responses are distributed for each of the names. Additionally, studies conducted with children have suggested that caution should be exercised when using a percentage of modal names alone as a name agreement measure (e.g., *Cannard et al., 2005*), and so this is usually complemented with a qualitative analysis of the children's responses (e.g., *Cycowicz et al., 1997*). One of the concerns is that, unlike adults, the name given by most children is not always identical to the expected name, and in some cases can even be incorrect.

For this reason, we complemented the measure with the percentage of participants who responded with alternative responses in different categories. Alternative names were classified within the categories developed by *O'Sullivan et al. (2012)*, that is, based on whether they were incorrect (e.g., physically similar such as *bombilla* [*bulb*] for *gota* [*drop*]), equivocal (i.e., non-existing word such as *rasca-uñas* [*scratch-nails*] for *lima* [*file*]), or correct (i.e., synonyms such as *vehículos* [*vehicles*] for *transporte* [*transport*]). Applying the same categorization rules described by *O'Sullivan et al. (2012)*, the classification of responses was made by two native Spanish judges and by a third judge when a consensus could not be reached.

The responses DKN, DKO, and TOT were excluded from the analysis of $H$ index, of percentage of modal name responses and of percentage of alternative responses because unknown responses were analyzed separately. The percentage of participants who provided the unknown responses in each category was calculated.

## Results

Table 2 provides descriptive statistics for the variables included in the analysis sorted by picture format: $H$ index, percentage of modal name responses, and percentage of alternative names in each category, percentage of DK responses in each category, familiarity, and visual complexity scores.

A Shapiro–Wilk test of normality was significant for the name agreement measures ($p < .001$), and non-significant for both subjective scales, familiarity ($W = 0.98$, $p = 0.272$), and visual complexity ($W = 0.99$, $p = 0.766$), indicating that data might be deviated from normality in name agreement measures but not in subjective scales. Figure 2
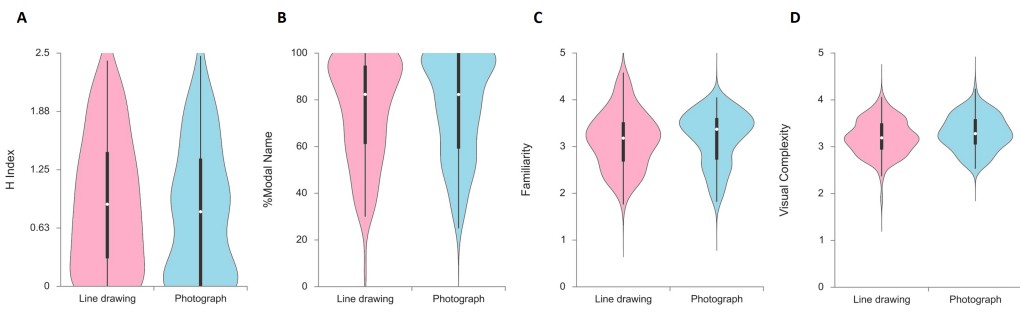

**Figure 2** Distribution of picture naming measures in Experiment 1, displayed as violin plots by picture format.

**Table 2** Means (and standard deviations) according to picture format in Experiment 1.

| Variable | Line drawing | Photograph |
|---|---|---|
| Name agreement | | |
| $H$ ind. | 0.90 (0.71) | 0.82 (0.71) |
| Modal name (%) | 75.47 (22.18) | 77.33 (21.52) |
| Alternative names | | |
| Incorrect names (%) | 14.30 (17.74) | 12.43 (15.92) |
| Equivocal names (%) | 0.85 (5.62) | 0.82 (4.56) |
| Correct names (%) | 9.38 (15.47) | 9.42 (15.46) |
| Unknown responses | | |
| Don't Know the Name (%) | 4.34 (8.56) | 5.38 (10.07) |
| Don't Know the Object (%) | 0.80 (3.07) | 1.16 (2.86) |
| Tip-Of-the-Tongue (%) | 0.25 (1.12) | 1.11 (5.05) |
| Subjective scales | | |
| Familiarity | 3.15 (0.61) | 3.19 (0.56) |
| Visual Complexity | 3.20 (0.37) | 3.29 (0.37) |

[1] In Experiment 1, we also analyzed the non-parametric alternative to the repeated measures ANOVA, the Friedman test, and the non-parametric alternative to $t$-test for paired samples, the Wilcoxon's signed rank test, and we found similar results. Significant differences were found between line drawings and photographs in $H$ index, $\chi^2 F(1) = 5.23$, $p = .022$, percentage of modal name, $\chi^2 F(1) = 5.55$, $p = .018$, percentage of incorrect alternative names, $W_s = 1262$, $p = .011$, and visual complexity, $W_s = 2053$, $p = .028$, but in contrast to the parametric tests, the differences were significant for the TOT responses, with higher percentage of TOT responses, $W_s = 4.5$, $p = .019$, in photographs than in line-drawings.

displays a violin plot of each picture naming variable for both formats. The white dot shows the median, the box includes the interquartile range, and whiskers are extended to the most extreme data point. Each side of the shaded line represents a kernel density estimation indicating the probability density of the data at different values. We constructed the plots using the free web-based tool Interactive Dotplot (*Weissgerber et al., 2017*).

In order to compare the name agreement measures between line drawing and photograph formats, we conducted mixed ANOVAs of $H$ Index and percentage of modal responses with Format (line drawing, photograph) as the within-subject factor and Order (line drawing—photograph, photograph—line drawing) as between-subjects factors. The analysis of variance (ANOVA) has proved to be robust when there is a deviation from the normality assumption,[1] and generally does not have strong effects on the Type I error rates or the power of the $F$-test (*Delacre et al., 2018*; *Harwell et al., 1992*; *Tiku, 1971*).

For the $H$ index, the significant main effect of format, $F(1,104) = 4.88$, $p = .029$, partial eta$^2 = 0.045$, indicated that values for line drawing were higher than those of the

photograph format. With respect to the between-subjects factors, the main effect of order, $F(1,104) = 0.31$, $p = .581$, partial $eta^2 = 0.003$, was not significant. Interaction between format and order, $F(1,104) = 1.02$, $p = .315$, partial $eta^2 = 0.010$, was not significant. These results suggest that picture format had an effect on name agreement. In particular, photographs decreased the dispersion of names given in comparison with line drawings, thus indicating that this format improved name agreement. This finding is in line with other similar evidence reported in adults (i.e., *Price & Humphreys, 1989*) and in young children employing different tasks other than naming, such as imitation or matching (*Callaghan, 2000*; *Ganea, Pickard & DeLoache, 2008*; *Simcock & DeLoache, 2006*).

For the percentage of modal responses, the main effect of format, $F(1,104) = 3.01$, $p = .086$, partial $eta^2 = 0.028$, and the main effect of order, $F(1,104) = 0.75$, $p = .390$, partial $eta^2 = 0.007$, were not significant. However, a significant interaction between format and order was found, $F(1,104) = 4.45$, $p = .037$, partial $eta^2 = 0.041$, indicating that the format manipulations had a differential impact as a function of whether photograph or line drawing was presented first or second. To further explore the format by order interaction, we carried out a separate dependent sample *t*-test for the drawing-photograph and photograph-drawing groups to compare format differences. In the drawing-photograph order, the significant difference of format, $t(52) = -2.45$, $p = .018$, $d = -0.68$, indicated that percentage of modal names for the photograph format ($M = 76.68\%$; $SD = 22.81$) was higher than that of the line drawing format ($M = 72.57\%$; $SD = 24.47$). In the photograph-drawing order, the percentage difference of modal responses between formats, $t(52) = 0.30$, $p = .764$, $d = 0.08$, was not significant. The reported statistical interaction revealed that, in modal name agreement, the format effect was likely to disappear when photographs were presented before line drawings. It is noteworthy to mention that while the effect of format for the $H$ index was reliable, the effect of format in the percentage of modal names failed to reach statistical significance in the photograph-drawing order. As we have already mentioned, the percentage of modal name responses is not sensitive to the dispersion of the responses as it only indicates the dominance of the name given by most subjects (e.g., *Snodgrass & Vanderwart, 1980*), and in children it should be interpreted with caution and complemented with other qualitative analyses (*Cycowicz et al., 1997*). Indeed, in our results children differed from the names intended by the experimenters (the word that the researchers used for the search and selection of the picture) nine of the 53 modal names (17%) in line drawings, and eight of 53 modal names (15%) in photographs. Of these, some of the modal names were even incorrect names such as *carta* [letter] for *sobre* [envelope], and *rana* [frog] for *sapo* [toad].

Given that the percentage of modal names provides limited information, particularly in samples with children (*Cannard et al., 2005*), we complemented the analyses by using the percentage of alternative names and the percentage of unknown responses. In order to examine differences between line drawings and photographs, a dependent sample *t*-test was conducted for each category respectively. For the alternative name responses, the dependent *t*-test revealed a significant difference in the mean percentage of incorrect names between formats, $t(105) = 2.39$, $p = .018$, $d = 0.47$, indicating that the average scores for line drawing were higher than those for photographs. The difference in mean

percentage scores between formats for equivocal names $t(105) = 0.09$, $p = .932$, $d = 0.18$, and correct names $t(105) = -0.05$, $p = .957$, $d = -0.01$, were not significant. As indicated by the alternative name analysis, stimuli presented as line drawings elicited similar correct and equivocal names but more incorrect names than stimuli presented in photograph format. These findings suggest that the surface and color information that is missing in line drawings increases visual ambiguity and evokes conceptual errors. For the unknown responses, the results revealed no significant difference between formats in terms of the average percentage of DKN responses, $t(105) = -1.78$, $p = .079$, $d = -0.35$, average percentage of DKO, $t(105) = -1.44$, $p = .153$, $d = -0.28$, and the average percentage of TOT, $t(105) = -1.76$, $p = .081$, $d = -0.34$, therefore indicating that picture format did not affect the percentage of unknown responses.

For the subjective scale measures, dependent t-tests revealed that the difference between the formats on the mean familiarity scores, $t(105) = -1.01$, $p = .315$, $d = -0.20$, was not significant, but there was a significant difference between formats in terms of the mean visual complexity scores, $t(105) = -2.45$, $p = .016$, $d = -0.48$, indicating that the average scores for photographs were higher than the average scores for line drawings. Photographic stimuli were rated as more visually complex, suggesting that children were sensitive to the greater number of details (such as surface details), presented by photographs in comparison with the line drawings.

## EXPERIMENT 2

Experiment 2 was conducted in order to replicate Experiment 1 using a different set of stimuli and a different sample, in order to ensure that the results were reliable and generalizable.

### Materials & methods
#### Participants
A total of 34 native Spanish-speaking children participated in the study: 19 were from the 3rd grade (50% girls, $M$ age = 8 years 4 months, $SD$ = 4.40 months) and 15 from the 4th grade (49% girls, $M$ age = 9 years 5 months, $SD$ = 5.10 months). None had received a diagnosis of neurological damage or problems with speaking or language. Three additional children were excluded from the sample for not completing the task. Details of the school, informed consent collection, and ethical approval were the same as in Experiment 1.

#### Materials
We used the 53 concepts of the PicPsy bank (N Martínez, H Matute & E Goikoetxea, 2019, unpublished data) that were not used in Experiment 1. Each concept depicted as both a photograph and a matched drawing delineated directly from the photograph (106 pictures total) was used.

We followed the same arrangement of the stimuli and we maintained equivalent properties in terms of the psycholinguistic variables and subjective ratings as in described in Experiment 1 (see Table 3). Like in Experiment 1, most of the concepts used in this experiment (approximately 79%) were taken from those reported by *Goikoetxea (2000)* for

**Table 3** Descriptive statistics of the psycholinguistic variables in Experiment 2.

| Variable | Database | M (SD) | Range |
|---|---|---|---|
| Linguistic variables | | | |
| Lexical frequency | *Martínez & García (2004)* | 148.52 (181.68) | 2.09–808.62 |
| Length | | 5.04 (1.70) | 2–10 |
| Subjective ratings | | | |
| Familiarity | *Duchon et al. (2013)* | 5.91 (0.72) | 3.57–6.91 |
| Imagination | *Duchon et al. (2013)* | 6.04 (0.58) | 4.86–6.75 |
| Concreteness | *Duchon et al. (2013)* | 5.87 (0.63) | 4.72–6.97 |
| Age of acquisition | *Alonso, Fernandez & Díez (2015)* | 4.31 (1.36) | 1.96–7.56 |

**Table 4** Means (and standard deviations) according to picture format in Experiment 2.

| Variable | Line drawing | Photograph |
|---|---|---|
| Name agreement | | |
| H ind. | 0.68 (0.64) | 0.55 (0.56) |
| Modal name (%) | 81.07 (21.02) | 83.96 (18.53) |
| Alternative names | | |
| Incorrect names (%) | 11.94 (16.49) | 9.21 (14.42) |
| Equivocal names (%) | 1.02 (9.01) | 0.19 (1.44) |
| Correct names (%) | 5.97 (11.00) | 6.64 (13.65) |
| Unknown responses | | |
| Don't Know the Name (%) | 1.80 (4.55) | 1.76 (4.03) |
| Don't Know the Object (%) | 1.54 (4.93) | 1.94 (4.67) |
| Tip-Of-the-Tongue (%) | 2.61 (4.20) | 1.53 (3.33) |
| Subjective scales | | |
| Familiarity | 3.37 (0.65) | 3.53 (0.83) |
| Visual Complexity | 3.51 (0.77) | 3.93 (0.86) |

3rd and 4th graders, and corresponded to 25 semantic categories (40% of them overlapped with those of Experiment 1). The 25 categories in the present experiment were: animals, atmospheric phenomena, birds, buildings, clothing, fruits, geographical accidents and natural land formations, kitchen utensils, light sources, mammals, media, parts of the human body, reading material, tools, trees, types of boats, types of building material, types of drinks, types of fabrics, types of food, types of plants, types of professions, types of relatives, types of toys, types of vehicles.

### Procedure and scoring

We followed the same procedure and scoring system as described in Experiment 1 and no significant differences were found in the age of the counterbalancing groups, $t(32) = -0.28$, $p = .782$, $d = -0.10$.

## Results

Table 4 provides descriptive statistics for the variables included in the analysis, sorted by format. A Shapiro–Wilk test of normality was significant for the name agreement measures

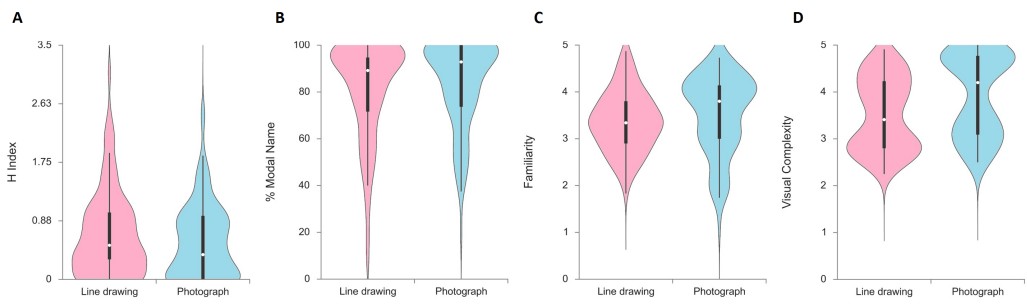

**Figure 3** Distribution of picture naming measures in Experiment 2, displayed as violin plots according to picture format.

(p <.001), and non-significant for the subjective scale of visual complexity ($W = 0.99$, $p = 0.539$), indicating that data might be deviated from normality in name agreement measures but not in subjective scales.

In order to compare the name agreement measures between line drawings and photographs, we conducted mixed ANOVAs[2] of $H$ Index and percentage of modal responses with Format (line drawings, photographs) as the within-subject factor and Order (line drawing—photograph, photograph—line drawing) as between-subject factors. Figure 3 displays a violin plot of each picture naming variable for both formats.

For the $H$ index, the significant main effect of format, $F(1,104) = 13.80$, $p = .000$, partial eta$^2$ = .117, indicated that values for line drawing were higher than those for the photograph format. With respect to the between-subject factors, the main effect of order, $F(1,104) = 0.75$, $p = .387$, partial eta$^2$ = 0.007, was not significant, whilst the interaction between format and order, $F(1,104) = 2.43$, $p = .122$, partial eta$^2$ = 0.023, also failed to reach significance. The replication of the format effect in the $H$ index is indicative of a higher name agreement for photographs in comparison with line drawings.

For the percentage of modal responses, a significant main effect of format, $F(1,104) = 7.07$, $p = .009$, partial eta$^2$ = 0.064, was found. With respect to the between-subjects factors, the main effect of order, $F(1,104) = 1.06$, $p = .306$, partial eta$^2$ = 0.010, was not significant. A significant interaction between format and order, $F(1,104) = 4.48$, $p = .037$, partial eta$^2$ = 0.041, was found. In the drawing-photograph order, the significant effect of format, $t(52) = -2.92$, $p = .005$, $d = -0.81$, indicated that the percentage of modal name responses for the photograph format ($M = 83.22\%$; $SD = 18.28$) was higher than those for the line drawing format ($M = 78.03\%$; $SD = 22.31$). In the photograph- drawing order, the difference between formats in terms of percentage modal responses, $t(52) = -0.47$, $p = .640$, $d = -0.13$, was not significant. Moreover, children differed from the names intended by the experimenters seven of the 53 modal names (13%) in line drawings, and five of 53 modal names (9%) in photographs. Of them, some of the modal names were even an incorrect name such as *papel* [paper] for *tela* [cloth], or *mujer* [woman] for *beso* [kiss]. As in Experiment 1, exploration of the significant interaction between format and order revealed that the effect of format was significant only in the drawing-photograph order. Previous studies have already documented greater naming accuracy for photographs

[2]After performing non-parametric analyses in Experiment 2, we found similar results. Significant differences were found between line drawings and photographs in $H$ index, $\chi^2 F(1) = 8.67$, $p = .003$, percentage of modal name, $\chi^2 F(1) = 6.67$, $p = .010$, percentage of incorrect alternative names, $W_s = 1166$, $p < .001$, percentage of TOT responses, $W_s = 404$, $p = .002$, and visual complexity, $W_s = 290.50$, $p < .001$.

compared with line drawings in adults (e.g., *Price & Humphreys, 1989*), suggesting that photographs provide information such as color and surface details, which can facilitate naming. Therefore, a possible explanation of the same interaction found in Experiment 1 and Experiment 2 could be that children's naming of a line drawing could be affected by the name given previously to the same stimulus presented in photograph format, reaching similar levels of name agreement in both conditions. However, children who named a photograph could be affected by the name given previously to the same stimulus in line drawing format, but the photograph provided them with more information to reach a significantly higher level of name agreement in photographs.

We complemented the analyses with the percentage of alternative names and the percentage of unknown responses in each category, respectively. For the alternative name responses, a dependent $t$-test revealed that the formats differed significantly in terms of the mean percentage of incorrect names, $t(105) = 3.21$, $p = .002$, $d = 0.63$, indicating that the average scores for line drawing were higher than those for photographs. The difference between formats in terms of mean percentage of equivocal names $t(105) = 1.10$, $p = .275$, $d = 0.21$, and percentage of correct names $t(105) = -0.93$, $p = .355$, $d = -0.18$, failed to reach significance. The higher percentage of incorrect names given to the line drawings in comparison with the photographs replicates the effect previously obtained in Experiment 1.

For the unknown responses, the results revealed that the formats did not differ significantly in terms of the average percentage of DKN responses, $t(105) = 0.12$, $p = .905$, $d = 0.02$, and average percentage of DKO, $t(105) = -1.64$, $p = .104$, $d = -0.32$. However, a significant difference between formats in the percentage of TOT responses, $t(105) = 2.78$, $p = .006$, $d = 0.54$, indicates that the average scores for line drawing were higher than those for photographs. In line with the findings of Experiment 1, the percentage of unknown responses was similar in both picture formats, except for the TOT category. However, unlike in Experiment 1, the percentage of unknown TOT responses was greater for line drawings than for photographs.

Finally, for the subjective scale measures, dependent t-tests revealed that the difference between the formats on the mean familiarity score, $t(105) = -1.42$, $p = .159$, $d = -0.28$, did not reach significance, whilst the formats differed significantly in terms of the mean visual complexity score, $t(105) = -11.17$, $p = .000$, $d = -2.18$, indicating that the average scores for photographs were higher than those for line drawings. As revealed by the results of Experiment 1, photographs were rated as familiar as line drawings, but visually more complex. As we noted in the introduction, line drawings are schematic and simple representations while photographs offer surface details that enrich the available visual information. Previous research with adults has shown that objective visual complexity is positively correlated with subjective visual complexity (*Shao & Stiegert, 2016*). Our results indicate that children subjectively rated the picture format to be more complex, a format that objectively incorporates more visual details.

## DISCUSSION

The purpose of the present experiments was to compare name agreement, alternative names, unknown responses, and familiarity and visual complexity measures between

photographs and matched line drawings in Spanish-speaking children. In Experiment 1, we found a general effect of picture format in children's naming accuracy and judgement of visual complexity. This was replicated in Experiment 2 using a different sample and a different picture set. Thus, we can summarize our main finding of the two experiments as showing higher name agreement and higher visual complexity values for the photograph format than for the line drawing format.

Photographs yield significantly more accurate naming performance than line drawings, specifically, less dispersion of name responses (*H* index), a higher percentage of children giving the modal name (only in line drawing—photograph order), and lower percentage of incorrect alternative names. Several factors might be important in explaining the advantage of photographs over line drawings in children's name agreement measures. For example, photographs offer surface details (e.g., texture, brightness, and shade,) and color information that line drawings do not, and this information facilitates recognition. Our results are in line with those from studies with adults showing better performance in object recognition (higher naming accuracy, lower response latency) for photographs— particularly colored photographs—than for line drawings of the same objects (e.g., *Brodie, Wallace & Sharrat, 1991*; *Price & Humphreys, 1989*). It has been suggested that color and surface cues activate more visual information than black-and-white pictures, and trigger semantic knowledge that facilitates object recognition. Another explanation for differences in perceptual processing that influence the retrieval of the concept can be related to the format of the two representations. For instance, *Uttl, Graf & Santacruz (2006)* suggested that a schematic representation of an object, such as a line drawing, might usually be perceived as a representation of a class of object or *type*, rather than as a representation of an individual object or *token*, like in the perception of photographs. Previous studies have provided support for the assumption of different sensory and perceptual processes for the identification of photographs and line drawings, such as strong embodiment and associations with real-world tangible objects (*Salmon, Matheson & McMullen, 2014*; *Saryazdi et al., 2018*). Further, similar findings in the previous literature have highlighted visual iconicity as one of the variables that particularly affects the performance of both children and adults. Previous research with young children showed the impact of visual iconicity on tasks involving the matching of pictures with real objects, with better performance for photographs than for line drawings (*Callaghan, 2000*; *Ganea, Pickard & DeLoache, 2008*). More recently, a subtle iconicity effect in picture naming has also been found in adults (*Saryazdi et al., 2018*). Our results in children using a picture-naming task showed that the more iconic representation (photograph) produced, the better name agreement measures were obtained, better than the measures of less iconic representation (line drawing).

Further, in the percentage of modal name responses, the format interacted significantly with order. The results of Experiment 1 and Experiment 2 indicate that in the photograph-drawing order, the effect of format was not significant. A possible explanation for this finding could be that most of the children recovered the name of the photograph when looking at the line drawing because it provided more information to facilitate naming. Previous evidence in adults (e.g., *Price & Humphreys, 1989*), and our results showing the

format effect in the $H$ index seems to support this explanation. The interaction found suggests that the modal name of a photograph is transferred quickly to the line drawing, a more schematic representation. In any case, the percentage of modal names could be an incomplete measure of name agreement in children (*Cannard et al., 2005*) and should be complemented with other quantitative and qualitative measures. Indeed, our complementary analysis of alternative responses revealed that line drawings elicited a statistically higher percentage of incorrect alternative names such as *piedra* [*rock*] for *nube* [*cloud*], *escalera [ladder]* for *cremallera [zipper]* or *abeja [bee]* for *mosca [fly]*.

The results from the subjective scales indicate that, broadly speaking, photographs were rated as being as familiar as the matched line drawings, but visually more complex. It is unsurprising that photographic stimuli were rated as more complex, given the greater details presented in comparison with the line drawings. In addition, objective and subjective visual complexity are positively correlated (*Shao & Stiegert, 2016*). Our results did not concur with those studies carried out with adults showing a similar or even higher visual complexity score for line drawings than for photographs (*Brodeur et al., 2010*; *Moreno-Martínez & Montoro, 2012*). Although it is possible that adults' ratings could be different from those given by children, we should interpret this finding with caution, given that these studies compare different samples and sets.

We believe that the results of this study have several implications for the selection of pictures as experimental and educational stimuli. As indicated by the differences in name agreement and visual complexity measures for photographs in comparison with line drawings, research on picture naming with children should pay attention to the type of stimuli employed, because this variable can generate differences in perceptual processing and influence the retrieval of the concept. On the one hand, line drawings have been widely used in research, assessment, educational, and child literature materials. Children are thus highly familiar with such stimuli. When children are exposed to simple drawings, they gather the typical salient elements of the object that help to develop figurative representations, flexibility of cognitive representation, and symbolic capacity during childhood (*Simcock & DeLoache, 2006*). This format also has less visual complexity, which could facilitate visual processing and influence naming latencies (*Shao & Stiegert, 2016*) whilst also having the added benefit of substantially reducing printing costs compared with colored photographs. On the other hand, colored photographs can facilitate object naming and provide ecological stimuli that allow specific transference to objects. This format can be an important vehicle for teaching vocabulary by virtue of a rich two-dimensional representation of real objects, particularly for those stimuli that are encountered less frequently in the child's environment. Current advancements in digital photography allow for color and surface details to be revealed, which could reduce the ambiguity of the presented real-life stimuli. Both line drawings and photographs are complementary instruments of excellent educational value since the exposure to a variety of pictures plays an important role in vocabulary acquisition.

## CONCLUSIONS

The current study aims at examining the effect of picture format in name agreement measures in two naming task experiments with children. The results showed that picture format affected name agreement measures, types of alternative names, and visual complexity, but not familiarity. These findings suggest that pictures were more likely to produce children's naming agreement and be rated as more visually complex when presented as photographs than when presented as line drawings. Importantly, the same results were observed using two different sets of stimuli in two different samples of children, which suggests that these are robust and generalizable results. Previous work found a picture format effect in adult's naming performance (e.g., *Price & Humphreys, 1989*) and in young children's non-verbal tasks such as matching pictures with real objects (e.g., *Callaghan, 2000*). The experiments presented here add evidence of the role of picture characteristics in children's naming to the existing literature, using large name agreement measures for each stimulus that contributes to a better understanding of the effect.

This work presents some limitations that could serve as a basis for new lines of research. In the case of our study, we matched the shape, size, orientation and contour of both formats but photographs recruited significant visual information such as surface details (such as texture, shade, three-dimensional cues), and particularly color information, that was not controlled. Therefore, in future work we recommend exploring possible format differences by adding colored drawings and black-and white-photographs to this dataset, but not grayscale line drawings because these stimuli provide less stable results (*Bonin et al., 2019*). Despite the challenges of data collection, it would be interesting to replicate this study in a sample of children younger than 8 years old. Previous research has shown that children under 8 years old are less efficient in picture naming than older children and adults (*Cannard et al., 2005*; *Cycowicz et al., 1997*), which could show stronger differences between formats or in different variables than those shown here. Moreover, the use of a repeated measure design where different variables are examined in two picture formats could affect judgements on the scales used, especially given the dependence between familiarity and visual complexity ratings (*Forsythe, Street & Helmy, 2017*). Finally, it would be interesting to address other measures such as naming speed, and examine a possible manipulability effect and other measures directly related to the visual characteristics of the image such as image agreement and image variability. Even though no previous work with children has examined these variables because they could be difficult tasks for primary school children to understand (*Wang, Chen & Zhu, 2014*), a recent meta-analysis in adults indicated that these two measures have a direct impact on picture processing (*Perret & Bonin, 2018*). Despite the mentioned limitations, we believe that future researchers and practitioners could benefit from the findings reported in these experiments, particularly when selecting visual stimuli to be used in naming tasks.

## ACKNOWLEDGEMENTS

We wish to thank Edurne Goikoetxea for critically reviewing the study proposal, and Aranzazu Viñas and Mario Álvarez for their help in the classification of alternative-names

responses. We also would like to express our gratitude to Carlos Sainz de los Terreros School for providing participants for the study and Sonia Cámara for helping in data collection.

### Funding
This work was supported by the Basque Government [Grant IT955-16]. The funders had no role in study design, data collection and analysis, decision to publish, or preparation of the manuscript.

### Grant Disclosures
The following grant information was disclosed by the authors:
Basque Government: IT955-16.

### Competing Interests
The authors declare there are no competing interests.

### Author Contributions
- Naroa Martínez conceived and designed the experiments, performed the experiments, analyzed the data, contributed reagents/materials/analysis tools, prepared figures and/or tables, authored or reviewed drafts of the paper, approved the final draft.
- Helena Matute conceived and designed the experiments, contributed reagents/materials/analysis tools, prepared figures and/or tables, authored or reviewed drafts of the paper, approved the final draft.

### Human Ethics
The following information was supplied relating to ethical approvals (i.e., approving body and any reference numbers):

The ethics committee of the University of Deusto approved the procedure of the present study (Ref: ETK-14/17-18).

### Data Availability
Data and materials are available at Open Science Framework: Naroa Martínez and Helena Matute. 2019. ''Data and Materials: Examining Differences of Picture Format in Children's Naming''. OSF. September 4. https://osf.io/d8cwp/https://osf.io/d8cwp/.

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
