# Peer review of "Examining the influence of picture format on children’s naming responses"

_PeerJ, doi:10.7717/peerj.7692_

## Round 0.1 · original submission · Major Revisions

· Academic Editor

Major Revisions

I have now received three review reports on your study, and have carefully read it myself. I thank the reviewers for their work. The assessment of the reviewers is convergent: they all think that this work is potentially of interest but raise a number of concerns that should be addressed in a revision.

The reviews are thoughtful and quite clear, so please refer to them for details. Notice that you do not necessary have to implement all the changes suggested; in case you do not, you should however clearly motivate your choice.

In the rest of my action letter let me focus on some issues that I consider crucial for a successful revision:

a. current literature - you should make an attempt at better grounding your work in current literature; the reviewers offer excellent suggestions you might want to follow (see comments of Reviewer 1 and 3);
b. choice of the sample - you should justify more clearly the choice of you r sample, offering a clear theoretical justification (see comments of Reviewer 3);
c. choice of the material and of the dimensions - you should provide theoretically funded justifications of the choice of your stimuli and dimensions to test. you might consider to introduce new dimensions to test and to expand the set of stimuli (see comments of Reviewer 2);
d. analyses. I agree with Reviewer 2 that a test of the normality of the distribution would be important to determine which kinds of analyses to perform.

Reviewer 1 ·

Basic reporting

see General comments for the author

Experimental design

see General comments for the author

Validity of the findings

see General comments for the author

Additional comments

Review of ms (#35015) entitled "Line drawing or photograph? Examining differences of picture format in children’s naming" by Naroa Martínez and Helena Matute for PeerJ

The manuscript reports "norms" that were collected from line-drawings and photographs on thirty-eight and thirty-four Spanish-speaking children aged from 8 to 10 years. The findings indicate that the format of pictures plays a role in the naming performance. Pictures of objects represented by photographs lead to higher name agreement scores (and less alternative incorrect names), higher visual complexity scores compared to the corresponding pictures of objects represented by line-drawings. The only exception is familiarity for which no significant difference was found between the two picture formats.
Evaluation
I found the study reported in the ms interesting. It addressed an important issue, namely whether certain psycholinguistic norms collected from line-drawings and from photographs of the same objects are different. In particular, different psycholinguistic "norms" were collected on children which is not often the case in the literature. In effect, most normative studies that have used pictures of objects have been conducted in adults. Therefore, the findings have important implications for the investigation of picture naming speed and accuracy in children. The methodology used is generally sound even though the choice of a complete within-design (Picture format x Norms) in the two studies is questionable because of carry-over effects. The analyses were correctly done. The overall message and the conclusions are clear. However, there are major concerns that need to be addressed before publication is concerned.

(1) The Introduction needs more details. Indeed, the review of the literature is clearly incomplete. There are a lot of normative studies that have been done on line-drawings or on photographs that are not reported in the ms but that are worth citing. Therefore, some good efforts should be made to include more of these studies in a revision. It is important to mention that a lot of studies have used the same sets of pictures (e.g., Snodgrass and Vanderwart, 1989 and the corresponding colorized version by Rossion and Pourtois, 2004) to collect norms in different languages and cultures. To give a few examples of studies that should be included in the ms:
*Navarrete, E., Arcara, G., Mondini, S., & Penolazzi, B. (2019). Italian norms and naming latencies for 357 high quality color images. PLoS ONE 14: e0209524.
*Bonin, P., Peereman, R., Malardier, N., Méot, A., & Chalard, M. (2003). A new set of 299 pictures for psycholinguistic studies: French norms for name agreement, image agreement, conceptual familiarity, visual complexity, image variability, age of acquisition, and naming latencies. Behavior Research Methods, Instruments, & Computers, 35, 158–167.
*Pind, J., Jónsdóttir, H., Tryggvadóttir, H.B., & Jónsson, F. (2000). Icelandic norms for the Snodgrass and Vanderwart (1980) pictures: Name and image agreement, familiarity, and age of acquisition. Scandinavian Journal of Psychology, 41, 41–48.
*Raman, I., Raman, E., & Mertan, B. (2014). A standardized set of 260 pictures for Turkish: norms of name and image agreement, age of acquisition, visual complexity, and conceptual familiarity. Behavior Research Methods, 46, 588–595.
*Russo, N., Hagmann, C. E., Andrews, R., Black, C., Silberman, M., & Shea, N. (2018). Validation of the C.A.R.E. stimulus set of 640 animal pictures: Name agreement and quality ratings. PLoS ONE, 13: e0192906.
*Tsaparina, D., Bonin, P., & Méot, A. (2011). Russian norms for name agreement, image agreement for the colorized version of the Snodgrass and Vanderwart pictures and age of acquisition, conceptual familiarity, and imageability scores for modal object names. Behavior Research Methods, 43, 1085–1099.

(2) Some references must be updated. I was very surprised to see that the authors were referring to one review paper published in 1996 (Johnson, Paivio, & Clark, 1996). Indeed, there are more recent reviews that the authors should be aware of and consider reporting in the revised version of their ms. Again, I provide the following two references that should be included in the revised ms, but there are certainly others worth mentioning, for instance certain chapters in Brenda Rapp’s (2000) book entitled "Handbook of Cognitive Neuropsychology: What Deficits Reveal About the Human Mind" (Psychology Press).
*Levelt, W. J. M., Roelofs, A., & Meyer, A. S. (1999). A theory of lexical access in speech production. Behavioral and Brain Sciences, 22, 1–75.
*Roelofs, A., & Ferreira, V. S. (in press). The architecture of speaking. In P. Hagoort (Ed.), Human language: From genes and brains to behavior. MIT Press

(3) There are two studies which are directly relevant to the issues addressed in the current study but that were surprisingly insufficiently described.
*The study by Bonin et al. (2019, Reading & Writing) is described only very slightly. Indeed, more space should be given to this study. In effect, the authors compared three different formats in written naming and investigated whether and how image characteristics influence written naming performance in adults. There were black-and-white pictures (Snodgrass & Vanderwart’s 1980 drawings), grayscale and colored pictures of the SV drawings as provided by Rossion and Pourtois (2004). Among the main findings, the authors found that colorized pictures yielded shorter written naming latencies than line drawings with the grayscale pictures being situated between the two. The addition of gray texture did not have an effect on the ease of recognition of the objects. The latter finding should be added more explicitly, for example by stating that the greyscale does not improve recognition as indexed by name-agreement scores. In any case, this study should be better described because it is fully relevant for the issues addressed in the ms.
* The methodology used by Saryazdi et al. et al. (2018) is interesting and should also be reported since it is share some features with the current study.

(4) There are also certain claims in the ms which are incorrect and should be corrected (see line-by-line comments).

Other line-by-line comments
Lines 48-51. I was really surprised to read that "Snodgrass and Vanderwart in their 1980 paper showed that pictures were named more readily when a particular word…" because in this publication the naming performance in adults was not measured online. The sentence must be reframed because this does not reflect was SV did exactly in their study.
Lines 52-53. Martein (1995) did not use the SV set of pictures.
Line 60. As far as MultiPic is concerned, please indicate some of the languages in which norms were collected in adults.
Lines 81-83. The authors mention "name agreement" and then "response latencies" when discussing the influence of picture format. This is not the same DV as they know. This must be corrected and clarified.
Lines 90-92. I do not understand what is the meaning of what is written here, please clarify.
Lines 94-96. The description of Johnson’s (1992) study is very confusing.
Lines 101-107. Please indicate explicitly which studies used black-and-white vs. colored drawings vs. black-and-white (or colored) photographs and compared performance in memory tasks and which compared performance in picture naming.
Lines 107-110. The description of Bramão et al. (2011) is very allusive. How was "recognition" measured in the 35 experiments?
Line 131. "name agreement, alternative names, unknown responses". When name agreement is measure using the H statistics, alternative names are taken into account. Please be explicit when describing the different measures.
Lines 151-187. Materials
More information concerning the stimuli set should be reported: the different semantic categories (and how many items per category). Instead of reporting in the main text the statistical characteristics of the items (Mean, SD, range) for the variables corresponding to frequency, length, etc., perhaps a table could be used instead and inserted in the Materials section.
Please indicate here whether the full set of pictures can be downloaded freely. This is an important information.
Lines 353-372. Same remark as for Experiment 1 (lines 151-187).
Discussion
Line 454-457. It should be noted that in adults the %NA agreement has been found to be higher with line drawings than with photographs. This should be mentioned somewhere in the ms and the conclusion about the influence of picture format on the naming performance should be more nuanced (lines 460-476). Indeed, photographs do not always facilitate the picture naming performance in adults even though in the present studies it was clearly observed that in children photographs yielded more accurate naming performance than line-drawings. For example, in adults the overall name agreement scores have been found to be higher in both the SV database (88%) and the Bonin et al. (2003) database (77%) than in the BOSS database (64% in the Brodeur et al. study [2010] and 59.45% with the latter pictures normed in Thai by Clarke, & Ludington, 2018).

·

Basic reporting

In the article entitled "Line drawing or photograph? Examining differences of picture format in children's naming", authors proposed to develop a photography database for picture naming experiments in children. They argued the choice of photographs rather than line drawing pictures because of the fact that several studies have reported an influence of format on both norms (e.g., Name Agreement) and spoken latencies. In general, the rational of the study is of good quality, the literature being well-cited and well-used. The subject treated is of great importance both for pscyhoeducation research (as indicated by authors) and also for fundamental researches. There are, however, certain points which prevent me from accepting this manuscript for publication in its present form. I do, however, show great interest in this study, and think that with modifications it could be a publication of importance.

Experimental design

A total of 38 children participated in the study. They are divided into two groups: one of 17 (3rd grade) and one of 21 (4th grade). A set of 53 concepts was used in both image and photographic format (106 stimuli in total). The material seems to me to correspond to what is classically used in this type of study.The authors asked the children to perform three tasks: one of written image naming, one of conceptual familiarity assessment and one of visual complexity. In this way, the authors obtained norms for the three factors: Name Agreement, Familiarity and Visual Complexity.

It is the same with regard to the calculation of different scores. Research question is weel defined and it is stated clearly how authors' research fills an identified knowledge gap. The study was conducted under the appropriate ethical conditions.The methodology seems to me quite suitable. There are enough details to allow replication.

Validity of the findings

There are two points in this part that ask me questions.
First, I'm not sure I understand the reason for doing two experiments. If the authors have (as I think) the goal of providing a standardized material for children's photography, there is no reason not to share the material to make one study. More generally, I'm not sure I understand the logic of using an experimental approach. Why make a standard collection study an experimental study?
Then, the authors use the analysis of variance to compare the normes obtained with the two image formats. I think that may be a problem. I am not sure that the reported data fit with a normal distribution. A test of Shapiro-Wilk or Kolmogorov-smirnoff can be used to test this point. If these tests are significant (as I think) a non-parametric tests (in this case, Mann-Withney rank tests) can be much more adequate and much less biais. Without this verification, it is hard to discuss results and conclusions.

Additional comments

Finally, I will have some general questions.
First, the authors worked on three classic norms: Name Agreement, Familiarity, and Visual Complexity. However, studies in adults indicate at least two other factors that have a direct impact on picture processing (for a recent synthesis see Perret & Bonin, 2018 Behav Res (2018) [url]Https://doi.org/[url] 10.3758 / s13428-018-1100-1): Image Agreement and Image Variability. Is there any particular reason why these factors (in direct relation to the visual characteristics of the image) have not been addressed?
Next, the authors used a relatively small sample of images, compared to other studies. Is there a particular reason for this small sample size?
In the same vein, are there criteria other than the linguistic ones that led to the selection of the data?
Finally, I am not sure what I mean by the phrase "No significant age differences were found ...".

I would like to reiterate the importance of this work to provide standardized photography material for children. It seems important, however, that authors address previous concerns before accepting the manuscript to publication.

Reviewer 3 ·

Basic reporting

As a preamble, please note that details concerning the different issues, that is to say basic reporting, experimental design, … are given in the attached file.
The submitted text generally complies with scientific articles standards. General structure is well organized. Bibliographical references are nearly sufficient, but are sometimes provided with erroneous or insufficiently precise and balanced arguments. Although the English is comprehensible, it seems sometimes uncomfortable and probably needs to be improved (as our proper English is sometimes questionable, we however do not try to give suggestions for this aspect). Results are however questionable essentially given the experimental design in use.

Experimental design

The issue addressed by this work pertains to the vast open debate concerning which type of 2-D images are better suited for such or such population and/or how and when cognitive processes could vary with the format. As pictures are sometimes the unique possibility to conduct experiments with children, for which the preceding questions were rarely addressed, it is a timely topic, which could be of interest for PeerJ. Despite the experimental design is in our opinion not well-suited and the materials choices are questionable, they are generally well described.

Validity of the findings

Participants and material choices are questionable given the literature and already existing databases : Arguments should be given to convince that the selected age band for participants is a good choice to speak about "children". Existing results indeed suggest that 8-10 years children productions are close to adults ones. The choice to build new sets of pictures while large normed databases already exist for adults should also be developed, particularly in view of name agreement measures, for which adults computations have been shown to be not well-suited for children.
The use of a repeated design leave uncertainty about the results that can be drawn from the data. We highly recommend to employ independent groups of participants, both for the norms and for the formats.
The way computations were done for one of the measured variables (NA%) should also be argued.
Bibliographical arguments given for supporting results have to be carefully checked in order to be more matched with the restricted measures that were done. They should be also more balanced.
These points are developed in details in the attach file.

Annotated reviews are not available for download in order to protect the identity of reviewers who chose to remain anonymous.

---

## Round 0.2 · accepted · Accept

· Academic Editor

Accept

I am happy to inform you that your paper has been accepted for publication on PeerJ.

# Reviewer 1 ·

Basic reporting

see general comments

Experimental design

see general comments

Validity of the findings

see general comments

Additional comments

This is my second review of this manuscript. The authors have taken seriously the concerns that I raised and the changes have resulted in a stronger paper. I have no remaining concern.

·

Basic reporting

OK All my concerns have been taken into account

Experimental design

OK. All my concerns have been taken into account

Validity of the findings

OK. All my concerns have been taken into account

Additional comments

OK. All my concerns have been taken into account. I think this manuscript is reaching for publication